## REPORT

# SCF-Fbxo42 promotes synaptonemal complex assembly by downregulating PP2A-B56

Pedro Barbosa[1], Liudmila Zhaunova[1], Simona Debilio[1,2], Verdiana Steccanella[1], Van Kelly[1], Tony Ly[1], and Hiroyuki Ohkura[1]

**Meiosis creates genetic diversity by recombination and segregation of chromosomes. The synaptonemal complex assembles during meiotic prophase I and assists faithful exchanges between homologous chromosomes, but how its assembly/ disassembly is regulated remains to be understood. Here, we report how two major posttranslational modifications, phosphorylation and ubiquitination, cooperate to promote synaptonemal complex assembly. We found that the ubiquitin ligase complex SCF is important for assembly and maintenance of the synaptonemal complex in *Drosophila* female meiosis. This function of SCF is mediated by two substrate-recognizing F-box proteins, Slmb/βTrcp and Fbxo42. SCF-Fbxo42 down-regulates the phosphatase subunit PP2A-B56, which is important for synaptonemal complex assembly and maintenance.**

## Introduction

Dynamic reorganization of the meiotic chromatin during prophase I is important for proper recombination and chromosome segregation. Formation of the synaptonemal complex is one of the canonical chromatin reorganization events. The synaptonemal complex is observed across species and promotes chromosome pairing and recombination (Page and Hawley, 2004). Moreover, assembly and disassembly of the synaptonemal complex need to be tightly regulated to ensure proper recombination and chromosome segregation. Although the molecular structure of the synaptonemal complex has been well characterized in recent years (Cahoon and Hawley, 2016), the molecular mechanism regulating assembly/disassembly of the synaptonemal complex remains poorly understood. A subsequent chromatin reorganization event in oocytes is the clustering of chromatin into a spherical structure called the karyosome/karyosphere (King, 1970; Parfenov et al., 1989). Its critical role in accurate chromosome segregation in oocytes has been shown (Cullen et al., 2005), but the molecular mechanisms regulating karyosome formation and function still remain to be understood.

Posttranslational modifications are thought to be key regulatory mechanisms of chromatin organization. Besides phosphorylation, ubiquitination is one of the most important posttranslational modifications in eukaryotes (Ciechanover et al., 2000). Ubiquitination consists of the addition of ubiquitin, a highly conserved 76-amino-acid polypeptide, to a wide range of proteins (Hershko and Ciechanover, 1998). Ubiquitination is most commonly known for targeting proteins for degradation by the proteasome complex (Hershko et al., 1982), but it can also change protein activity, localization, and protein–protein interactions. Importantly, defects in the

ubiquitination system are associated with the development of many human diseases, including neurodegenerative diseases, autoimmunity, and cancer (Ciechanover and Schwartz, 2004). Ubiquitination of a substrate results from a multistep mechanism mediated by three classes of enzymes: ubiquitin-activating enzymes (E1s), ubiquitin-conjugating enzymes (E2s), and ubiquitin ligases (E3s; Pickart and Eddins, 2004). A large number of predicted E3 ubiquitin ligases are encoded in genomes (600 in humans and 207 in *Drosophila*; Du et al., 2011; Medvar et al., 2016), but for most of them, their substrates and biological function are unknown.

The SCF (Skp1–Cul1–F box) complex is one of the best characterized ubiquitin ligases and is a key regulator of the cell cycle. It adds ubiquitin to protein substrates, which are then degraded by proteasomes (Deshaies, 1999). It consists of three core subunits (Roc, Cul1, and Skp1) and one variable subunit (an F-box protein; Willems et al., 2004). As an F-box protein directly recognizes the substrate, different F-box proteins are thought to confer different substrate specificities to SCF (Bai et al., 1996). Therefore, SCF consists of multiple complexes with distinct substrate specificity rather than a single homogenous complex. Although a number of F-box proteins have been identified in each organism (69 in humans and 45 in *Drosophila*; Kipreos and Pagano, 2000), only few of them have had substrates or biological functions identified. This suggests that we are far from understanding the full range of functions executed by SCF.

Here, we report that SCF ubiquitin ligase is important for synaptonemal complex assembly in *Drosophila* female meiosis. We identify two F-box proteins, Fbxo42 and Slmb/βTrcp, that mediate regulation of the synaptonemal complex. Our

[1]Wellcome Centre for Cell Biology, School of Biological Sciences, University of Edinburgh, Edinburgh, UK; [2]Department of Biology and Biotechnology, University of Pavia, Pavia, Italy.

Correspondence to Hiroyuki Ohkura: h.ohkura@ed.ac.uk.



biochemical and genetic evidence suggests that Fbxo42 promotes synaptonemal complex assembly by negatively controlling the level of the protein phosphatase PP2A-B56.

## Results and discussion

### SkpA is important for formation and maintenance of the synaptonemal complex

To identify genes involved in chromosome organization in *Drosophila* female meiosis (Fig. 1 A), we performed a targeted screen of predicted ubiquitin-associated enzymes. We found that knockdown of the *SkpA* gene by RNAi in ovaries resulted in abnormal karyosome morphology. *SkpA* is one of eight *Skp1* homologues in *Drosophila* (Murphy, 2003; Du et al., 2011). In contrast to a spherical morphology seen in controls, RNAi of *SkpA* resulted in distorted karyosome morphology (Fig. 1, B and C). We did not observe other defects often reported alongside abnormal karyosome morphology in other mutations or RNAi. These defects absent in *SkpA* RNAi include extensive attachment to the nuclear envelope (Breuer and Ohkura, 2015; Lancaster et al., 2007), persistent DNA double-stranded breaks detected by γH2Av, or meiotic recombination checkpoint dependency assessed using a mutation of the checkpoint kinase Mnk/Chk2 (Staeva-Vieira et al., 2003; Fig. S1, A and B).

To test whether SkpA depletion affects the synaptonemal complex, immunostaining of its main component, the transverse protein C(3)G, was performed (Fig. 1, D and E). Meiotic progression during oogenesis is sequentially observed in regions of the germarium (regions 2a, 2b, and 3) and in stage 2–14 oocytes in a well-defined pattern (Fig. 1 A; King, 1970; Lake and Hawley, 2012). In control meiotic cells, the synaptonemal complex was assembled as a filamentous structure in region 2a of the germarium (Fig. 1, D and E), as previously reported (Page and Hawley, 2001). This filamentous structure was maintained in the oocyte until stages 5–7, when it gradually disassembled (Page and Hawley, 2001). In contrast, assembly of the filamentous synaptonemal complex was greatly reduced in SkpA-depleted oocytes (Fig. 1, D and E). Furthermore, even when the filamentous synaptonemal complex was assembled, it was prematurely disassembled by region 3, much earlier than in the control. A possibility of off-target effects was excluded by expression of a wild-type RNAi-resistant *SkpA* that rescued the karyosome and synaptonemal complex defects (Fig. S1, C and D). Therefore, SkpA is required for proper karyosome morphology and assembly/maintenance of the synaptonemal complex.

The synaptonemal complex along the arms and at centromeres are differentially regulated by two meiotic cohesin complexes (the C(2)M complex and the Ord complex), respectively (Gyuricza et al., 2016). Coimmunostaining for the centromere protein CenpA/Cid and C(3)G in SkpA-depleted oocytes showed that the synaptonemal complex was prematurely disassembled from chromosome arms earlier than the control but maintained at centromeres until late stages as in the control (Fig. S2 A). Furthermore, SkpA depletion greatly reduced recruitment of the C(2)M complex to chromosomes (Fig. S2 B). Therefore, SkpA is required for recruiting the C(2)M complex, which is responsible for synaptonemal complex assembly along chromosome arms.

### SkpA functions as part of the SCF complex

The SCF consists of four subunits: Skp1, Cul1, Roc, and an F-box protein (Willems et al., 2004; Fig. 2 C). The *Drosophila* genome encodes 8 Skp1 homologues, 7 Cul1 homologues, 3 Roc homologues, and 45 F-box proteins (Du et al., 2011). Theoretically, many combinations of SCF subunits are possible. To define the exact SCF composition involved in the synaptonemal complex regulation or karyosome formation, SkpA fused with GFP (GFP-SkpA) was expressed and immunoprecipitated from dissected ovaries using an anti-GFP nanobody. Ovaries expressing GFP alone were used as a comparison. Quantitative mass spectrometry revealed Cul1, Roc1a, and 15 F-box proteins specifically coimmunoprecipitated with SkpA (Fig. 2 A).

To test whether these subunits are indeed required for regulation of the synaptonemal complex or karyosome formation, Cul1 was first depleted by RNAi. Immunostaining of C(3)G showed that the synaptonemal complex was assembled later and disassembled earlier in meiotic cells partially depleted of Cul1 (Fig. 2, D and E), similar to *SkpA* RNAi. The karyosome morphology was also defective (Fig. S2 C). However, RNAi of Roc1a using three different shRNAs did not show defects (Table S1), which may be due to insufficient depletion or functional partial redundancy among Roc homologues (Donaldson et al., 2004). These results showed that an SCF complex containing SkpA and Cul1 regulates the synaptonemal complex and the karyosome.

The SCF complex is known to be regulated by the neddylation cycle, the dynamic addition and removal of Nedd8 (Pan et al., 2004; Reitsma et al., 2017). Depletion of the key regulators Cand1 and CSN5 disrupted the assembly of the filamentous synaptonemal complex and karyosome formation (Fig. S3, A and B), showing that the neddylation cycle is important in these processes. This further confirms the essential role of the SCF complex in regulating the synaptonemal complex and the karyosome.

### Two substrate-recognizing F-box proteins, Slmb/βTrcp and CG6758/Fbxo42, mediate regulation of the synaptonemal complex

By SkpA coimmunoprecipitation, we also identified 15 F-box proteins that form complexes with SkpA in ovaries (Fig. 2 A). Each SCF complex contains only one F-box protein, which acts as the substrate-recognizing subunit. Therefore, it is crucial to determine which F-box protein is required for regulating the synaptonemal complex. To determine this, we generated new transgenic lines expressing shRNA against each of the 15 F-box proteins. Each of the SkpA-interacting F-box proteins was depleted by RNAi, and oocytes were immunostained for C(3)G. Among the 15 F-box proteins, single depletion of two F-box proteins (Slmb and CG6758) showed an incomplete formation and/or premature disassembly of the synaptonemal complex (Fig. 3, A and B; Table S1). These two F-box proteins are nonredundantly required for promoting the synaptonemal complex, suggesting that SCF is involved in multiple pathways.

Slmb is well conserved and the orthologue of human βTrcp. Slmb/βTrcp is one of the best-studied F-box proteins (Fuchs et al., 1999; Jiang and Struhl, 1998; Spevak et al., 1993; Suzuki et al., 1999; Yaron et al., 1998). Several substrates of Slmb/βTrcp have been identified, including Wee1, Cdc25A, Emi1, and Plk4

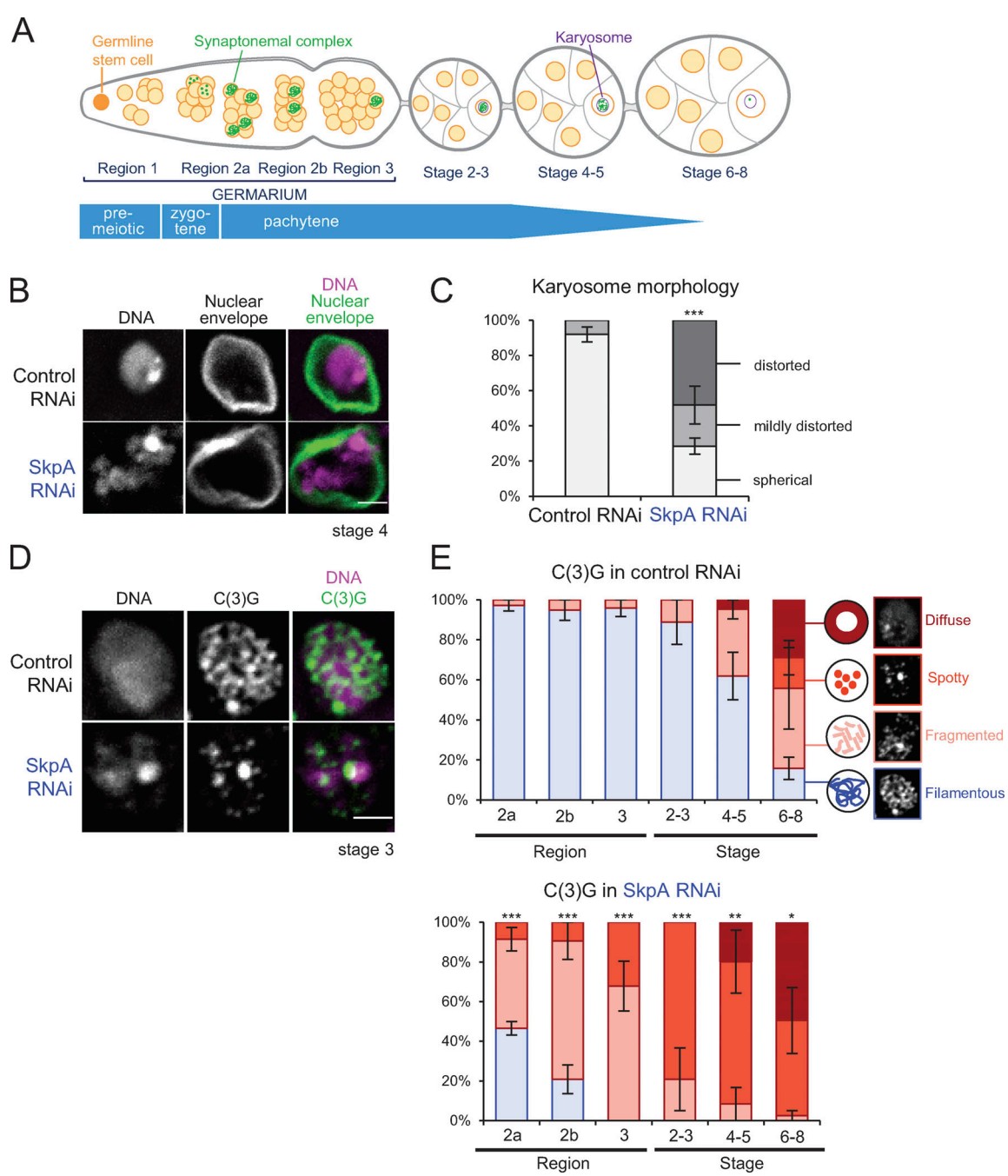

Figure 1. **SkpA is important for assembly and maintenance of the synaptonemal complex. (A)** Meiotic progression in *Drosophila* ovaries. **(B)** DNA and the nuclear envelope protein Lamin in control and *SkpA* RNAi oocytes. Scale bar = 2 μm. ControlRNAi, *nos-GAL4(MVD1)/UASp-shRNA(white)*; SkpA RNAi, *nos-GAL4(MVD1)/UASp-shRNA(SkpA)*. **(C)** The karyosome morphology in control and *SkpA* RNAi oocytes. Error bars represent SEM from triplicated experiments representing 24–75 oocytes for each stage. ***, P < 0.001. **(D)** The synaptonemal complex component C(3)G and DNA in control and *SkpA* RNAi oocytes. Scale bar = 2 μm. **(E)** The synaptonemal complex morphologies (C(3)G localization patterns) in control and *SkpA* RNAi through meiotic progression. Error bars represent SEM from triplicated experiments representing 10–31 germaria/oocytes for each region/stage. *, P < 0.05; **, P < 0.01; ***, P < 0.001.

(Busino et al., 2003; Guderian et al., 2010; Margottin-Goguet et al., 2003; Watanabe et al., 2004), although none of these substrates has been linked to the synaptonemal complex or the karyosome in any species. Slmb/βTrcp depletion resulted in impaired assembly and premature disassembly of the synaptonemal complex, as well as abnormal karyosome morphologies (Fig. 3, A and B; and Fig. S2 C). Expression of another shRNA against Slmb/

βTrcp resulted in similar synaptonemal complex and karyosome defects, indicating that an off-target effect is unlikely.

In addition, we identified a newly characterized F-box protein, CG6758, as a key regulator of the synaptonemal complex and the karyosome. CG6758 is conserved and the orthologue of human Fbxo42, but studies of Fbxo42 orthologues have been very limited in any species. Depletion of CG6758 (which we call

Figure 2. **An SCF complex containing SkpA and Cul1 regulates the synaptonemal complex. (A)** A volcano plot of proteins immunoprecipitated with GFP-SkpA or GFP from ovaries. For each protein, the mean ratio of signal intensities found in immunoprecipitates of GFP-SkpA and GFP alone was plotted against the statistical significance (the P value given by *t* test) from biological triplicates. **(B)** A list of proteins immunoprecipitated specifically with GFP-SkpA. **(C)** A diagram of the SCF complex and interacting proteins. **(D)** The synaptonemal complex component C(3)G and DNA in control and *Cul1* RNAi. Both control and *Cul1* RNAi were induced by a weaker driver (*MVD2*) in contrast to all other RNAi (*MVD1*). Scale bar = 2 μm. **(E)** C(3)G localization patterns in control and *Cul1* RNAi throughout meiotic progression. Error bars represent SEM from triplicated experiments representing 14–33 germaria/oocytes for each region/stage. *, P < 0.05.

Fbxo42) prevented assembly of the fully filamentous synaptonemal complex and prematurely disassembled the partially formed structure (Fig. 3, A and B), as well as disrupting karyosome formation (Fig. S2 D). Expression of two other shRNAs against Fbxo42 led to similar defects, demonstrating that these defects are caused by knockdown of this gene rather than off-target effects.

Altogether, we found that two SCF complexes that consist of SkpA–Cul1-Slimb/βTrcp (SCF-Slimb/βTrcp) and SkpA–Cul1–Fbxo42 (SCF-Fbxo42) are important for proper formation and

maintenance of the synaptonemal complex and karyosome formation in *Drosophila* female meiosis.

### Fbxo42 decreases the PP2A-B56 Wrd protein level to promote synaptonemal complex assembly

An F-box protein provides substrate specificity to SCF through direct interaction with substrates. To identify substrates of SCF containing Fbxo42 or Slimb/βTrcp (SCF-Fbxo42 or SCF-Slimb/βTrcp), GFP-tagged Fbxo42 or Slimb/βTrcp was expressed and

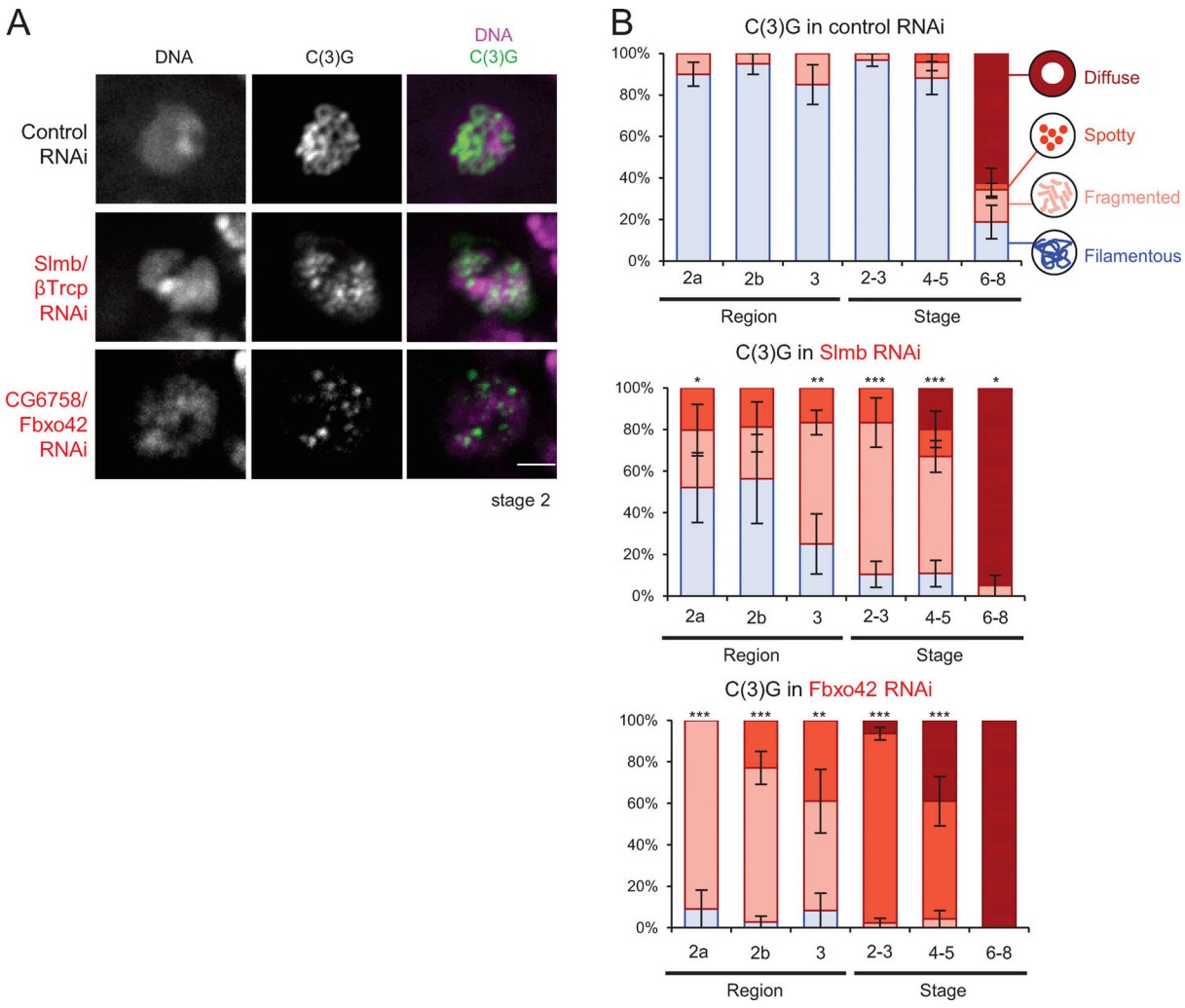

**Figure 3.** **SCF complexes containing Slmb/βTrcp or CG6758/Fbxo42 mediate regulation of the synaptonemal complex. (A)** The synaptonemal complex component C(3)G and DNA in control, *Slmb/βTrcp*, and *Fbxo42* RNAi throughout meiotic progression. Scale bar = 2 µm. **(B)** C(3)G localization patterns in control, *Slmb/βTrcp* RNAi, and *Fbxo42* RNAi. Error bars represent SEM from triplicated experiments representing 20–31 germaria/oocytes for each region/stage. *, P < 0.05; **, P < 0.01; ***, P < 0.001.

immunoprecipitated from dissected ovaries using an anti-GFP nanobody. Using GFP-expressing ovaries as a control, quantitative mass spectrometry was performed. In addition to SCF subunits, 4 and 37 proteins were coimmunoprecipitated specifically with Fbxo42 (Fig. 4 A) and Slmb/βTrcp (Fig. S3 D), respectively. We decided to focus on these four Fbxo42-interacting proteins: the catalytic (C) subunit and the structural (A) subunit of protein phosphatase 2A (PP2A) and the two 14–3-3 isoforms (14–3-3ε and ζ).

We showed that SCF-Fbxo42 regulates synaptonemal complex assembly and karyosome formation. Because SCF is typically known to ubiquitinate substrates for degradation by the 26S proteasome (Cardozo and Pagano, 2004), we hypothesized that one or more substrates of SCF-Fbxo42 need to be destroyed for proper assembly of the synaptonemal complex and karyosome formation. We predicted that overexpression of this critical substrate in wild-type would phenocopy the synaptonemal complex and/or karyosome defects of Fbxo42 depletion. As an F-box protein is the substrate-recognition subunit (Petroski and

Deshaies, 2005), the Fbxo42-interacting proteins we identified are good candidates for SCF-Fbxo42 substrates.

To test whether any of these are critical substrates, we generated transgenic flies expressing GFP-tagged 14–3-3 isoforms and PP2A subunits (C and A) that interact with Fbxo42. In addition, we also generated transgenic flies that can express various B regulatory subunits of PP2A, including Tws (B; B55), Wrd (B′; B56), and Wdb (B′; B56), as the B subunits confer substrate specificity to the core PP2A complex (C + A; Chen et al., 2007). These proteins were individually overexpressed in otherwise wild-type ovaries, and the synaptonemal complex and the karyosome were examined by immunostaining. We found that overexpression of GFP-tagged Wrd, one of two alternative B56 subunits of PP2A, resulted in delayed assembly and premature disassembly of the synaptonemal complex, as well as abnormal karyosome morphologies (Fig. 4, B and C; and Fig. S2 E), as seen in Fbxo42 RNAi. In contrast, overexpression of the other proteins did not affect the synaptonemal complex or the karyosome, although we cannot exclude the possibility that they are still

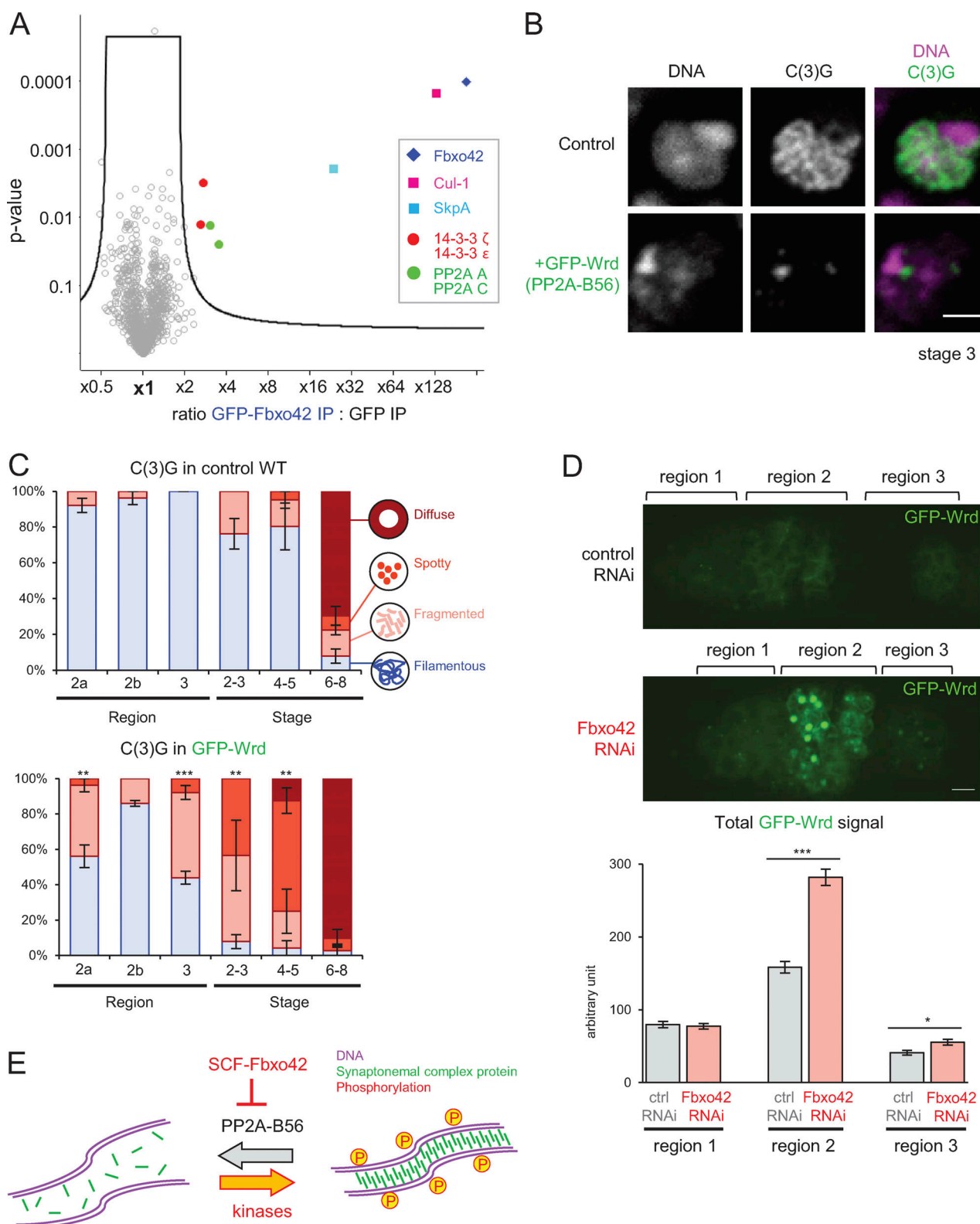

Figure 4. **Fbxo42 depletion increases PP2A-B56 (Wrd) levels, which destabilize the synaptonemal complex. (A)** A volcano plot of proteins immunoprecipitated with GFP-Fbxo42 or GFP from ovaries. For each protein, the mean ratio of signal intensities found in immunoprecipitates of GFP-Fbxo42 and GFP alone was plotted against the statistical significance (the P value given by *t* test) from biological triplicates. SCF subunits, PP2A core subunits and 14–3-3 proteins were significantly enriched in GFP-Fbxo42 immunoprecipitates. **(B)** Localization of the synaptonemal complex component C(3)G in meiotic cells expressing GFP-Wrd (PP2A-B56) and a control. Scale bar = 3 μm. GFP-Wrd, *nos-GAL4(MVD1) Wrd⁺/UASp-GFP-Wrd Wrd⁺*; control, *nos-GAL4(MVD1) Wrd⁺/Wrd⁺*. **(C)** C(3)G localization patterns in meiotic cells overexpressing GFP-Wrd (PP2A-B56) and control cells throughout meiotic progression. Error bars represent SEM

from triplicated experiments representing 19–24 germaria/oocytes for each region/stage. **(D)** GFP-Wrd signals in control and Fbxo42 RNAi live germaria. The images were captured and processed using identical conditions to allow accurate comparison. Total GFP signals above the background signals were quantified in regions 1, 2, and 3 of germaria (zygotene to early pachytene) expressing GFP-Wrd with control or *Fbxo42* shRNA. Error bars represent SEM from analysis of 32–54 germaria for each. Scale bar = 2 µm. Fbxo42 RNAi, *nos-GAL4(MVD1) Wrd⁺/UASp-GFP-Wrd Wrd⁺ UASp-shRNA(FBxo42)*; control RNAi, *nos-GAL4(MVD1) Wrd⁺/UASp-GFP-Wrd Wrd⁺ UASp-shRNA(white)*. **(E)** A schematic model showing that SCF-Fbxo42 down-regulates PP2A-B56 to tip the balance of phosphorylation toward synaptonemal complex assembly. *, P < 0.05; **, P < 0.01; ***, P < 0.001.

regulated by SCF (Table S1). Thus, restricting the PP2A-B56 Wrd level is critical for proper regulation of the synaptonemal complex.

If SCF-Fbxo42 was responsible for targeting Wrd for degradation, then Fbxo42 depletion would increase the Wrd protein level during oogenesis. To test this possibility, Fbxo42 was depleted in ovaries expressing GFP-Wrd, and the protein level of GFP-Wrd was estimated by GFP fluorescence in live ovaries (Fig. 4 D). GFP-Wrd without Fbxo42 depletion showed weak cytoplasmic signals at all stages. In the Fbxo42 depletion, GFP-Wrd showed significant increases in region 2 of the germarium (equivalent to the zygotene to pachytene stages; Fig. 4 D). These results demonstrated that SCF-Fbxo42 down-regulates the protein level of the PP2A-B56 Wrd during oogenesis.

### SCF-Fbxo42 stabilizes the synaptonemal complex by down-regulating PP2A-B56

Here, we first showed that SCF activity is important in early meiotic stages for assembly and maintenance of the synaptonemal complex. This itself represents a significant discovery. Recently, the mouse Skp1 was shown to be important for maintaining the synaptonemal complex in spermatocytes (Guan et al., 2020), suggesting that this role of SCF is conserved. We also showed that SCF depletion disrupts recruitment of a C(2)M meiotic cohesin complex and karyosome formation, suggesting underlying defects in meiotic chromosome organization. SCF consists of three core subunits and one variable F-box protein that recognizes specific substrates (Cardozo and Pagano, 2004). A number of F-box proteins have been identified (45 in *Drosophila*; Du et al., 2011), but only few have had their substrates or biological functions identified. Therefore, our identification of two F-box proteins, Fbxo42 and Slmb/βTrcp, which mediate this SCF function in meiosis, represents an important advance in the field. In particular, our identification of the conserved but previously uncharacterized Fbxo42 is an unanticipated finding.

It is crucial to identify downstream targets of ubiquitin ligases, but the transient nature of ubiquitination makes it technically challenging. We successfully showed that SCF-Fbxo42 down-regulates the phosphatase PP2A-B56 to promote synaptonemal complex assembly (Fig. 4 E). Involvement of protein kinases, including the chromosomal passenger complex (Aurora B kinase) and Plk1, in synaptonemal complex assembly has been well documented in various species (Argunhan et al., 2017; Chen et al., 2007; Jordan et al., 2012; Sourirajan and Lichten, 2008), but the importance of regulating protein phosphatases is much less recognized. Our results show that in addition to high-kinase activities, it is crucial to keep phosphatase activity low, and this is achieved by SCF-Fbxo42 down-regulating the protein level of the phosphatase PP2A-B56. Our study revealed a direct

cooperation between two major posttranslational modifications, phosphorylation and ubiquitination, in synaptonemal complex assembly.

## Materials and methods
### *Drosophila* handling and techniques
Standard fly techniques were followed according to Ashburner et al. (2005). The *Drosophila* stocks used were cultured at 25°C with standard cornmeal media. Young female flies were matured with males at 25°C (2 d) or 18°C (4–6 d) for immunostaining of early stages. For RNAi in ovaries, a GAL4 driver line, *MVD1 (P {GAL4::VP16-nos.UTR}CG6325^{MVD1}* BDSC4937; Bloomington Drosophila Stock Center), was crossed with RNAi lines we generated and the following Transgenic RNAi Project (TRiP) RNAi lines (Harvard Medical School): *SkpA* RNAi (HMS00791; BDSC32991), *Cand1* RNAi line 1 (GL00445; BDSC35605), *Roc1a* (HMS00353; BDSC32362), and *Nedd8* (HMS00818; BDSC33881). A weaker driver, *MVD2 (P{GAL4::VP16-nos.UTR}MVD2; BDSC 7303)*, was used for *Cul1* RNAi (GL00561; BDSC36601) and *CSN5* RNAi (HMJ30047; BDSC62970), as *MVD1* gave very small ovaries. The phenotype of each RNAi was compared with control RNAi (*white* RNAi; GL00094; BDSC35573) using the same driver. *MVD1* was used to express a GFP-tagged protein under the *UASp* promoter.

To generate transgenic fly lines expressing GFP-SkpA, GFP-CG6758/Fbxo42, GFP-Slmb, GFP-C(2)M, or GFP-tagged PP2A subunits, phiC31 integrase-mediated transgenesis onto the third chromosome was performed by Best Genes using the *VK33* line (BDSC9750), which carries an *attP* site at 65B2 on the third chromosome. Similarly, to generate transgenic fly lines expressing shRNA against the F-box proteins, the line (BDSC8622) carrying an *attP* site at 68A4 on the third chromosome (*attP2*) was used for phiC31 integrase-mediate transgenesis. Flies expressing an RNAi resistant wild-type *SkpA* transgene were generated by P-element–mediated transgenesis performed by Best Genes.

The meiotic recombination checkpoint was suppressed by a heterozygous mutation, *mnk^{p6}* (DmChk2; Klattenhoff et al., 2007), and *spnA* RNAi (GL00669; BDSC38898) was used as a positive control.

### Molecular techniques
Standard molecular techniques were followed throughout (Sambrook et al., 1989). Expression plasmids were generated using the Gateway cloning system (Invitrogen). To generate entry plasmids, the coding sequences or the coding sequences including intervening introns of the genes were first PCR amplified from either cDNA or genomic DNA. The amplified

sequence with a stop codon and the additional GAAA before the initiation codon was introduced between the NotI and AscI sites of pENTR/D-TOPO (Invitrogen) using Gibson assembly (New England Biolabs). For *c(2)M*, the coding sequence with a stop codon and the additional CCAAA before the initiation codon was recombined with pDONR221 (Invitrogen) by Gateway BP Clonase enzyme (Invitrogen). The following cDNAs provided by the Drosophila Genomics Resource Center (Indiana University, IN) were used as templates for PCR: *SkpA* (HL01263), *slmb* (LD08669), *c(2)M* (GM03132) *wrd* (LD29902), *wdb* (LD34343), *tws* (LD12394), and *mts* (LD26077). For *CG6758/Fbxo42* and *PP2A-29B*, genomic DNA from *w^{III8}* was used as PCR templates.

The coding sequences were then recombined into Gateway destinations vectors using LR Clonase II following the manufacturer's protocol (Invitrogen). To generate transgenic flies expressing a GFP-tagged protein under the *UASp* promoter, φPGW modified from destination vector pPGW of the Murphy's Gateway collection by adding the φC31 *attB* recombination site was used (Beaven, 2017). Transgenic flies expressing GFP-SkpA, GFP-Slmb, GFP-CG6758, GFP-C(2)M, and GFP-tagged PP2A subunits (Wrd, Wdb, mts, PP2A-29B, and Tws) generated by this method were used in this study.

To generate a wild-type *SkpA* transgene resistant to shRNA (HMS00791), three silent mutations (5′-ACGCAA**A**AC**G**TT**T**AAC ATTAA-3′; the mutations are underlined and in bold) were introduced into the region of *SkpA* that is recognized by the shRNA. These mutations were introduced by PCR-amplifying the pENTR containing the SkpA-coding sequence using overlapping primers with the desired mutations and assembled using Gibson assembly (New England Biolabs). The pENTR containing shRNA-resistant version of *SkpA* was inserted into the destination vector containing a ubiquitin promoter pUW (the pUbi plasmid converted into a destination vector) by LR Clonase II enzyme (Invitrogen).

To generate flies expressing shRNA, the following design of two complementary oligonucleotides with cohesive ends were inserted between the EcoRI and NheI sites of the Walium22 plasmid by ligation using the TRiP protocol (https://hwpi.harvard. edu/files/fly/files/2ndgenprotocol.pdf?m=1465918000).

First, 21-nt sense sequences were selected from the exon sequence using an online tool DSIR (http://biodev.extra.cea.fr/DSIR/DSIR.html) based on Vert et al. (2006). Each of these 21-nt sense sequences was reverse complemented without a 2-nt offset to generate the 21-nt antisense sequence. Using these two sequences, the two following oligonucleotides were designed: 5′-CTAGCAGT + (21-nt sense sequence) + TAGTTATATTCAAGCATA + (21-nt antisense sequence) + GCG-3′ and 5′-AATTCGCGGAGC + (21-nt sense sequence)+TATGCTTGAATATAACTA + (21-nt antisense sequence) + ACTG-3′.

For *CG6758/Fbxo42*, 5′-GGAGCACTATTATCCTTTATT-3′ at position 493 of the coding sequence was selected as the sense sequence. For *slmb*, 5′-GGTGCGCAAGAAAGACTCATC-3′ at position 573 of the coding sequence was selected. The sequences used to generate the remaining RNAi constructs are listed in Table S1.

## Immunostaining of karyosome in *Drosophila* oocytes

For immunostaining of karyosome at early stages, 0- to 24-h-old females flies were matured with males at 25°C for 2 d on standard cornmeal media with an excess of dried yeast. Ovaries from five flies were dissected in 200 µl Robb's medium (100 mM Hepes, pH = 7.4, 55 mM sodium acetate, 40 mM potassium acetate, 100 mM sucrose, 10 mM glucose, 1.2 mM MgCl₂, and 1 mM CaCl₂). After dissection oocytes were fixed in 100 µl of karyosome fixative buffer (8% formaldehyde [Sigma-Aldrich], 100 mM cacodylate, pH 7.2, 100 mM sucrose, 40 mM potassium acetate, pH 7.5, 40 mM sodium acetate, and 10 mM EGTA) for 10 min. After fixation, oocytes were washed in 200 µl PBS with 0.1% Triton X-100 and further incubated with 200 µl PBS with 10% FBS and 1% Triton X-100 for 2 h. After blocking, the oocytes were washed for 10 min in 200 µl PBS with 0.1% Triton X-100 and incubated overnight in 100 µl primary antibody solution (primary antibodies diluted in 100 µl PBS with 0.1% Triton X-100). After incubation in primary antibody, oocytes were washed three times for 10 min each with PBS with 0.1% Triton X-100. After washing, the oocytes were incubated for 4 h with 100 µl secondary antibody solution (secondary antibodies diluted in 100 µl PBS with 0.1% Triton X-100 and containing DAPI at 0.4 µg/ml). Oocytes were then washed three times for 10 min in washing buffer and mounted on glass microscope slides in 85% glycerol/2.5% propyl gallate.

The following primary antibodies and dilution factors were used: mouse monoclonal anti-Lamin antibody (1/100; ADL67.10; Developmental Studies Hybridoma Bank), rat anti-C(3)G antibody (1/100; Zhaunova et al., 2016), rabbit polyclonal anti-GFP antibody (1/100; Life Technologies). Secondary antibodies conjugated with Cy3, Alexa Fluor 488, or Cy5 (Jackson Laboratories) were used at a 1/250 dilution. DNA was stained using DAPI (Sigma-Aldrich) at 0.4 µg/ml concentration.

## Immunostaining of the synaptonemal complex in *Drosophila* oocytes

0- to 24-h-old female flies were matured for 2 d at 25°C in a vial on freshly yeasted media in the presence of males. Ovaries from five flies were dissected in cold 1x PBS. After dissection, oocytes were fixed for 20 min in 800 µl fixative solution (containing 200 µl PBS with 2% formaldehyde [Sigma-Aldrich], 0.5% Triton X-100, and 600 µl heptane [BDH Chemicals]). After fixation, oocytes were washed in 200 µl of PBS with 0.1% Triton X-100 and further incubated with 200 µl of blocking solution (PBS with 10% FBS and 1% Triton X-100) for 2 h. After blocking, oocytes were washed for 10 min in 200 µl PBS with 0.1% Triton X-100 and incubated overnight in 100 µl primary antibody solution (primary antibodies diluted in PBS with 0.1% Triton X-100). After incubation in primary antibody, oocytes were washed three times for 10 min in PBS with 0.1% Triton X-100. After washing, oocytes were incubated with 100 µl secondary antibody solution (secondary antibodies diluted in PBS with 0.1% Triton X-100 and containing DAPI at 0.4 µg/ml) for 4 h. Oocytes were then washed three times for 10 min in PBS with 0.1% Triton X-100 and mounted on glass microscope slides in a 85% glycerol/2.5% propyl gallate solution.

The following primary antibodies and dilution factors were used: rabbit anti-Cid antibody (1/800; Active Motif), rabbit anti-γH2Av antibody (1/100; Lancaster et al., 2010), rat anti-C(3)G antibody (1/100; Zhaunova et al., 2016), and rabbit polyclonal

anti-GFP antibody (1/100; Life Technologies). Secondary antibodies conjugated with Cy3, Alexa Fluor 488, or Cy5 (Jackson Laboratories) were used at a 1/250 dilution. DNA was stained using DAPI (Sigma-Aldrich) at 0.4 µg/ml concentration.

## Image analysis of fixed ovaries

Immunostained oocytes were imaged with a confocal scan head LSM800 attached to an Axiovert 200M (Zeiss) using Plan-ApoChromat objective lens (63×/1.4 numerical aperture) with Immersol 518F oil (Zeiss). Z-sections were captured with 0.5-µm interval, 512 × 512-pixel/zoom 2 (~0.1 µm/pixel). The maximum intensity projection of multiple Z-planes covering the region of interest is shown in the figures. Contrast and brightness were adjusted uniformly across the field using ImageJ. Images were exported as a tagged image file (TIFF) and edited using ImageJ.

The karyosome morphologies were classified as spherical when the karyosome showed a spherical shape, mildly distorted when spherical shape was mostly maintained, and distorted when spherical shape was largely disrupted. For the quantification of the karyosome, oocytes between stages 4 and 6 were scored. A two-tailed $t$ test was used for statistical analysis to compare the percentage of spherical karyosomes between control and experimental RNAi.

The C(3)G localization pattern in meiotic prophase was classified into four patterns: "filamentous" (showing relatively long filamentous structures that represent fully assembled synaptonemal complex), "fragmented" (showing very short filamentous structures that represent the synaptonemal complex starting to disassemble or in the process of assembling), "spotty" (showing a small number of strong C(3)G foci in the nuclei), and "diffuse" (showing a C(3)G signal that is diffused evenly in the nucleoplasm, representing complete disassembly). For quantification of synaptonemal complex morphology, region 2a was classified for each germarium based on the majority of the C(3)G staining pattern in multiple nuclei that accumulated C(3)G. For region 2b, where two nuclei accumulate C(3)G, each germarium was scored according to the morphology of the better-formed synaptonemal complex. For region 3, where only one nuclei accumulates C(3)G, each germarium was quantified according to the patterns previously mentioned.

For each region or stage, at least 15 germaria or oocytes from five or more flies were scored for C(3)G staining patterns. The experiments were triplicated to obtain the mean percentages and SEMs for each pattern. A two-tailed $t$ test was used for statistical analysis comparing percentages of filamentous nuclei between control and RNAi oocytes, except $SkpA$ shRNA rescue, for which percentages of the filamentous C(3)G pattern were compared between $SkpA$ shRNA meiotic cells and $SkpA$ shRNA meiotic cells expressing an shRNA-resistant wild-type $SkpA$ transgene. The actual P values (from left to right) are as follows: 0.00008 (Fig. 1 C); 0.0001, 0.0006, 0.000001, 0.0002, 0.0015, and 0.019 (Fig. 1 E); 0.026, 0.21, 0.11, 0.024, 0.53, and 0.85 (Fig. 2 E); 0.028, 0.059, 0.002, 0.0000005, 0.0003, and 0.044 (Slmb; Fig. 3 B); 0.0003, 0.000004, 0.0009, 0.0000001, 0.00003, and 0.059 (Fbxo42; Fig. 3 B); 0.0088, 0.06, 0.0001, 0,0019, 0.0052, and 0.35 (Fig. 4 C); 0.0036 (Fig. S1 C); 0.00027, 0.0010, 0.00069, 0.00014, 0.07, and 1.0 (Fig. S1 D); 0.03 (Fig. S2 C); 0.0014 and 0.0014 (Fig. S2 D); 0.00048 (Fig. S2 E); 0.0010, 0.0026, 0.31, 0.14, 0.079, and 0.20 (Cand1; Fig. S3 B); 0.0025, 0.0030, 0.003, 0.0006, 0.13, and 0.37 (CSN5; Fig. S3 B); and 0.016 and 0.012 (Fig. S3 C).

## Live imaging

0- to 24-h-old female flies were matured for 2 d at 25°C in a vial on freshly yeasted media in the presence of males. The ovaries were dissected at room temperature in a drop of Halocarbon 700 oil (Halocarbon) on a coverslip (24 × 50 mm). Early stage oocytes expressing GFP-tagged Wrd were imaged under a microscope (Axiovert; Zeiss) attached to a spinning disk confocal head (CSU-X1; Yokogawa) controlled by Volocity (PerkinElmer). A Plan-Apochromat objective lens (63×/1.4 numerical aperture) was used with Immersol 518F oil (Zeiss).

Crosses generating $Fbxo42$ RNAi and control flies expressing GFP-Wrd were set up at the same time, and germaria for both genotypes were analyzed in parallel. A total of 32–54 germaria from 8–10 adult females from three independent crosses were used for each genotype. To quantify the total GFP-Wrd signal intensity, the maximum intensity projection was first performed for a three-dimensional image of each germarium. The total GFP signal intensity (S) was measured in a square (size = a) drawn around each of the regions corresponding to region 1, 2, or 3; the average background signal (b) was then estimated from a square drawn in an area without ovaries. The total GFP signal intensity above the background was estimated as S – (a × b). A two-tailed $t$ test was used to compare the average total GFP signal intensity above the background. The actual P values in regions 1, 2, and 3 (Fig. 4 D) are 0.69, $10^{-12}$, and 0.011, respectively.

## Immunoprecipitation of GFP-tagged SCF components from *Drosophila* ovaries

Ovaries from transgenic flies expressing GFP-SkpA (Fig. 2 A), GFP-CG6758/Fbxo42 (Fig. 4 A), GFP-Slmb/βTrcp (Fig. S3 D) or GFP-alone were dissected at room temperature in 1x PBS. 60 flies from each genotype were dissected and frozen with a minimum carryover of the buffer on a 1.5-ml tube prechilled on dry ice. Collected ovaries were snap frozen in liquid nitrogen and stored at –80°C.

For immunoprecipitation, ovaries were transferred to 400 µl homogenization buffer (20 mM Tris-HCl, pH 7.5, 50 mM NaCl, 5 mM EGTA, and 1 mM DTT supplemented with 1 mM PMSF, protease inhibitors [one tablet/10 ml cOmplete, Mini EDTA-free; Roche], 1 µM okadaic acid, 10 mM p-nitrophenyl phosphate, and 0.5% Triton X-100) on ice. Ovaries were homogenized with a 1 ml glass Dounce homogenizer and incubated on ice for 30 min. Lysed samples were transferred into a 1.5-ml tube and centrifuged at 13,000 rpm for 30 min at 4°C. 400 µl supernatant (input) was incubated for 30 min at 4°C with 30 µl GFP-MA trap beads (Chromotek), which were previously washed three times in wash buffer (20 mM Tris-HCl, pH 7.5, 50 mM NaCl, 5 mM EGTA, and 0.1% Triton X-100). After 30 min of incubation at 4°C on a rotating wheel, the beads were washed twice in cold homogenization buffer containing 0.1% Triton X-100. During the last washing step, beads were transferred to a new 1.5-ml tube

and washed with homogenization buffer without Triton X-100 and kept at −20°C. As a control, ovaries expressing GFP alone were processed in parallel alongside ovaries expressing a GFP-tagged SCF subunit.

### Mass spectrometry from immunoprecipitation of GFP-tagged SCF components

To prepare samples for mass spectrometry, 2% SDS was added into the samples followed by boiling at 100°C for 5 min. The boiled samples were centrifuged at 15,000 *g* for 10 min. The supernatant was transferred to a new tube, and 25 mM of 0.5 M Tris (2-carboxyethyl) phosphine (TCEP; Thermo Fisher Scientific) and 25 mM of 0.5 M Iodoacetamide (IAA; Sigma-Aldrich) were added; the tube was left for 1 h in the dark at room temperature. After 1 h, 4 vol of 100% acetone was added to the samples and left overnight at 4°C. The next day, the samples were centrifuged at 15,000 *g* for 10 min, the supernatant was removed, and the pellet washed with 500 µl acetone for 1 h at −20°C. Centrifugation and the acetone wash were repeated one more time, followed by one more centrifugation and wash with 90% ethanol for 1 h. The samples were centrifuged once more at 15,000 *g* for 10 min and the supernatant discarded. The pellet was resuspended with 200 µl 100 mM of Triethylammonium bicarbonate (TEAB; Sigma-Aldrich; from 1 M stock). The peptides were digested with 0.5 µg of trypsin (Thermo Fisher Scientific) at 37°C overnight. The next day, the samples were digested for an extra 6 h with an additional 0.5 µg trypsin. Finally, samples were desalted with C18 stage tips. The tips were first conditioned with 50 µl 100% acetonitrile followed by incubation twice with 150 µl of 0.1% formic acid to equilibrate the column. The samples were loaded into the stage tips previously acidified with 1% formic acid. Then, the tips were washed twice with 150 µl of 0.1% formic acid. The peptides were eluted from the stage tips with 25 µl of 2:1 acetonitrile/0.1% formic acid, dried, and resuspended in 0.1% Trifluoroacetic acid (TFA) for liquid chromatography mass spectrometry analysis.

An Ultimate 3000 RSLCnano HPLC (Thermo Fisher Scientific) was coupled via electrospray ionization to an Orbitrap Elite Hybrid Ion Trap-Orbitrap (Thermo Fisher Scientific). Peptides were loaded directly onto a 75-µm × 50-cm PepMap-C18 EASY-Spray LC Column (Thermo Fisher Scientific) and eluted at 250 nl/min using 0.1% formic acid (solvent A) and 80% acetonitrile/0.1% formic acid (solvent B). Samples were eluted over a 120-min stepped linear gradient from 1% to 30% B over 105 min, then to 42% B over an additional 15 min, followed by wash and equilibration. MS1 scans of 1E6 ions were acquired in the Orbitrap at 60,000 resolution settings over 350–1,700 m/z and with a 445.120025 lock mass. This was followed by 20 data-dependent MS2 collision induced dissociation (CID) events (5E3 target ion accumulation) in the ion trap at rapid resolution with a 2 D isolation width, a normalized collision energy of 35, 50 ms maximum fill time, a requirement of a 1E4 precursor intensity, and a charge of ≥2. Precursors within 5 ppm were dynamically excluded for 40 s after two repeats.

Data were processed using MaxQuant version 1.6.2.6 (www.nature.com/articles/nbt.1511) and searched against *Drosophila melanogaster* UniProt up000000803 proteome sequences, allowing for variable methionine oxidation and protein N-terminal acetylation and fixed cysteine carbamidomethylation. A target-decoy threshold of 1% was set for both peptide-spectrum match (PSM) and protein false discovery rate. Match-between-runs was enabled with identification transfer within a 0.7-min window.

The MaxQuant Proteingroups.txt file was loaded into Perseus version 1.6.2.1; potential contaminants, reversed sequences, and proteins that were only identified by site were removed from the dataset. The label-free quantitation (LFQ) intensities were transformed into $\log_2$ to achieve normal distribution, which was verified by visual inspection of histogram distribution plots of $\log_2$-transformed data generated in Perseus for each sample. Only proteins identified in at least three runs were considered for LFQ and entries with an LFQ equal to zero were kept.

Statistical significance of changes in abundance between sample groups (GFP-alone versus GFP-SkpA, GFP-alone versus GFP-CG6758/Fbxo42, and GFP-alone versus GFP-Slmb/βTrcp) was calculated by a two-tailed *t* test (P < 0.01). The results were filtered using the Benjamini–Hochberg procedure for false discovery rate correction (false discovery rate <0.05).

The mass spectrometry proteomics data have been deposited to the ProteomeXchange Consortium via the PRIDE (Perez-Riverol et al., 2019) partner repository with the dataset identifier PXD022755.

### Online supplemental material

Fig. S1 shows that SkpA depletion does not affect the timing of DSB formation and repair and that the karyosome defect is independent of the meiotic recombination checkpoint. Fig. S2 shows that SkpA is required for recruiting the atypical C(2)M cohesin complex along chromosome arms. Fig. S3 shows the importance of the neddylation cycle and Slmb/βTrcp interacting proteins. Table S1 lists phenotypes of lines expressing shRNA or a GFP-tagged protein.

## Acknowledgments

We are grateful to the members of the Ohkura laboratory for their support and discussion.

The Bloomington Drosophila Stock Center/Resource Center (National Institutes of Health grants P40OD018537 and 2P40OD010949-10A1) and the Transgenic RNAi Project at Harvard Medical School (National Institutes of Health/National Institute of General Medical Sciences grant R01-GM084947) provided fly stocks and reagents. This work is supported by the Wellcome Trust (grants 081849, 098030, 206315, 099827, 092076, 203149, and 206211). P. Barbosa and T. Ly are supported by The Darwin Trust Scholarship and a Royal Society/Wellcome Sir Henry Dale Fellowship, respectively.

The authors declare no competing financial interests.

Author contributions: P. Barbosa designed and performed experiments, analyzed the data, and wrote the manuscript. L. Zhaunova, S. Debilio, V. Steccanella, and V. Kelly performed experiments and analyzed the data. H. Ohkura and T. Ly designed experiments and wrote the manuscript.

Submitted: 24 September 2020

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

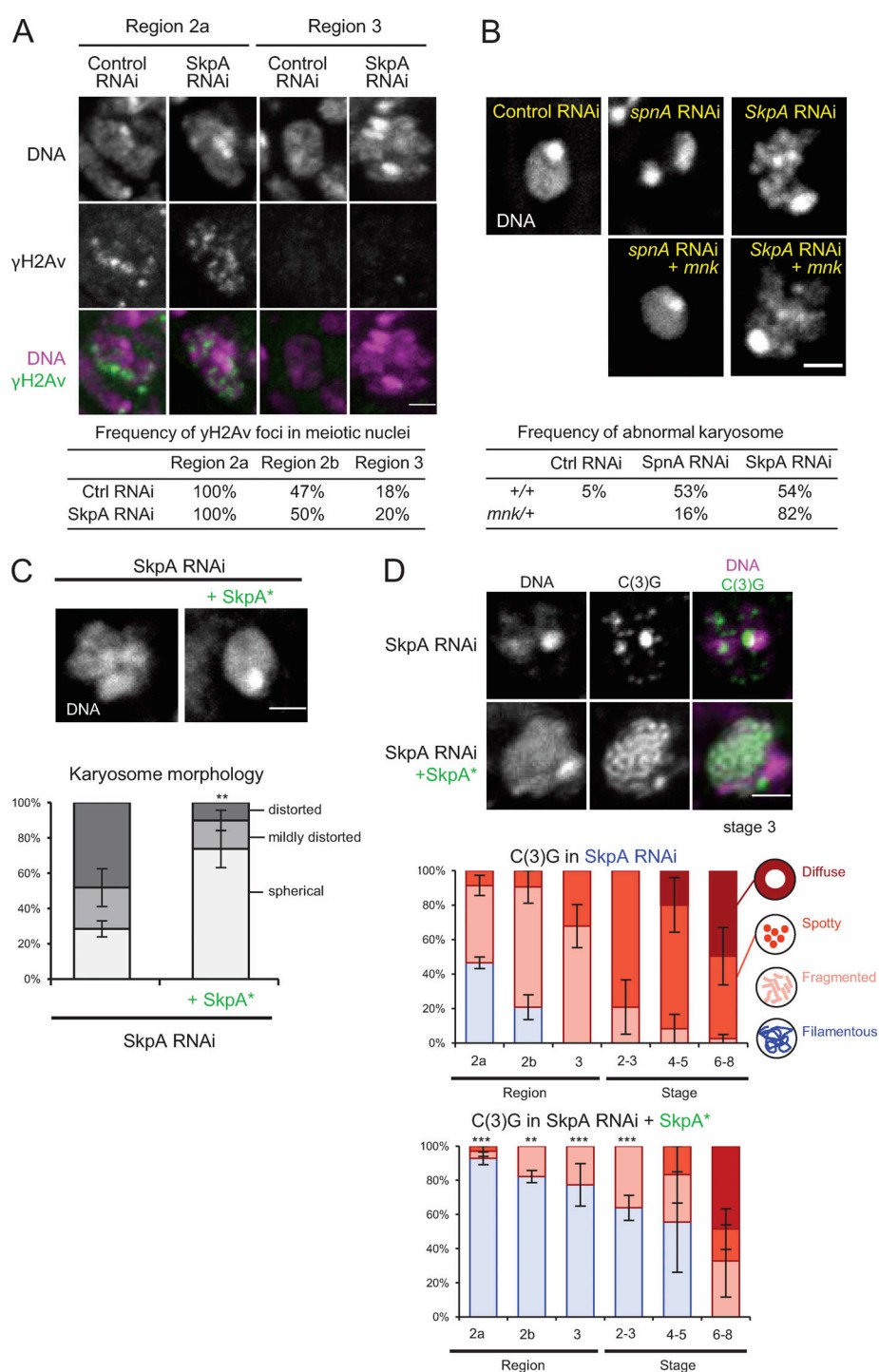

Figure S1. **SkpA depletion does not affect the timing of DSB formation and repair, and the karyosome defect is independent of the meiotic recombination checkpoint. (A)** Immunostaining of γH2Av, which marks DNA double-strand breaks alongside DNA staining, and frequencies of γH2Av foci in meiotic nuclei in control and *SkpA* RNAi. Meiotic cells were identified by costaining of C(3)G. A total of 14–17 germaria were counted for each region. **(B)** DNA staining of the karyosomes in *spnA* and *SkpA* RNAi oocytes with or without a heterozygous *mnk (Chk2)* mutation, and frequencies of abnormal karyosome morphology. The karyosome abnormality is independent of the meiotic recombination checkpoint. Enhancement of karyosome defects by a *mnk* mutation is common and not specific to SkpA or SCF. A total of 19–39 oocytes were counted for each genotype. Control RNAi, *nos-GAL4(MVD1)/UASp-shRNA(white)*; *spnA* RNAi, *nos-GAL4(MVD1)/UASp-shRNA(spnA)*; *SkpA* RNAi, *nos-GAL4(MVD1)/UASp-shRNA(SkpA)*; *spnA* RNAi +*mnk*, *nos-GAL4(MVD1) mnk^p6/UASp-shRNA(spnA)*; *SkpA* RNAi +*mnk*, *nos-GAL4(MVD1) mnk^p6/UASp-shRNA(SkpA)*. **(C)** DNA staining of karyosomes in *SkpA* RNAi oocytes with or without a wild-type RNAi-resistant *SkpA* transgene (SkpA*) expressed under the ubiquitin promoter, and frequencies of karyosome morphologies. Scale bar = 2 µm. Error bars represent SEM from triplicated experiments representing 49–52 oocytes for each genotype. SkpA RNAi, *nos-GAL4(MVD1)/UASp-shRNA(SkpA)*; SkpA RNAi +SkpA*, *nos-GAL4(MVD1) Ub-SkpA(RNAi resistant)/UASp-shRNA(SkpA)*. **(D)** Localization patterns of the synaptonemal complex component C(3)G in *SkpA* RNAi with or without a wild-type RNAi-resistant *SkpA* transgene. Scale bars = 2 µm. Error bars represent SEM from triplicated experiments representing a total of 11–23 germaria/oocytes for each region/stage. **, $P < 0.01$; ***, $P < 0.001$.

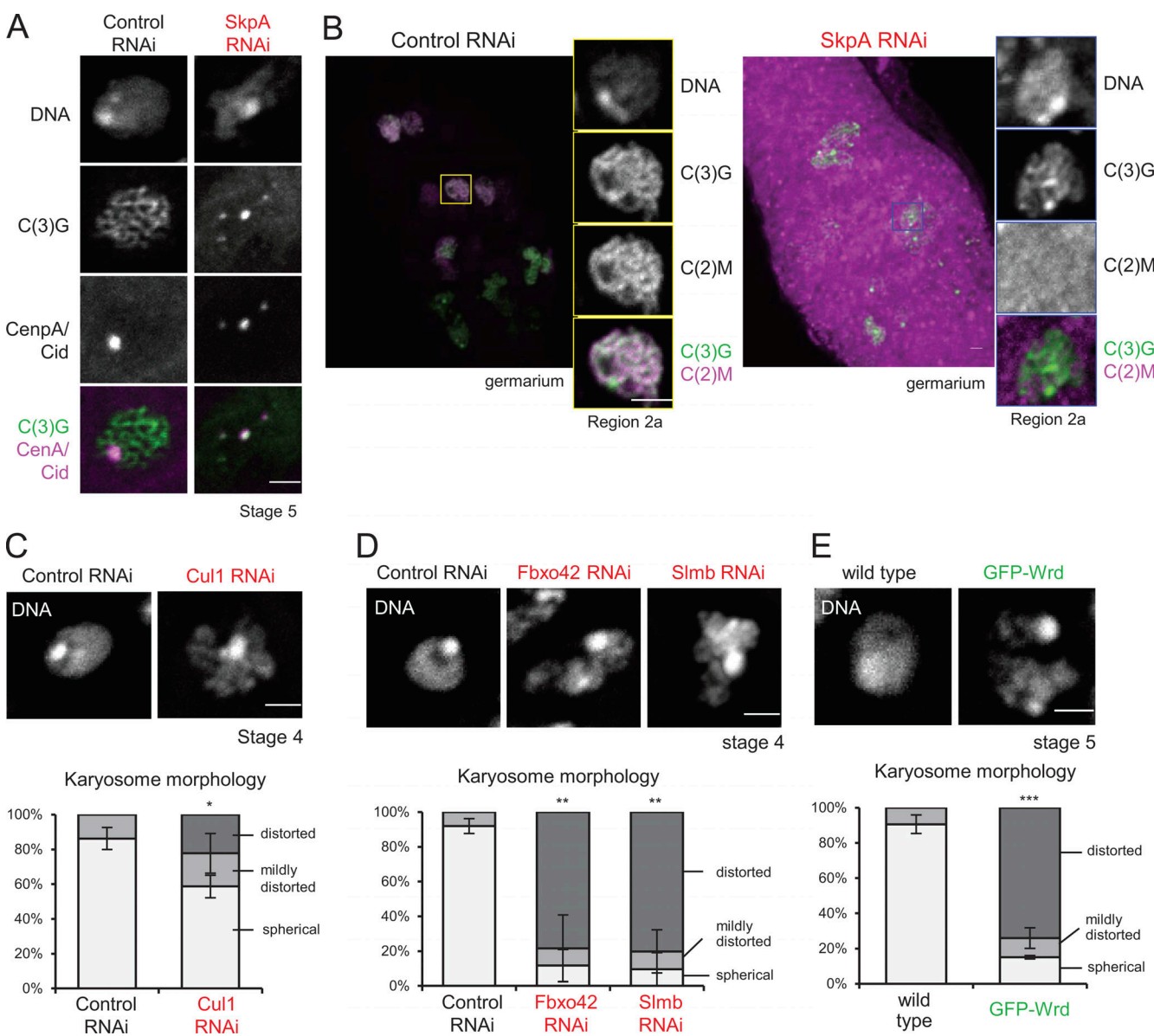

Figure S2. **SCF depletion results in defects in meiotic chromosome organization. (A)** SkpA is required for maintenance of the synaptonemal complex along chromosome arms, but not centromeres. The synaptonemal complex component C(3)G and the centromere protein CenpA/Cid localizations in control and *SkpA* RNAi oocytes. C(3)G and CenpA/Cid were immunostained in ovaries expressing shRNA against *white* (control) and *SkpA* genes. The synaptonemal complex was disassembled except at centromeres in *SkpA* RNAi oocytes. Scale bar = 2 μm. SkpA RNAi, *nos-GAL4(MVD1)/UASp-shRNA(SkpA)*; control RNAi, *nos-GAL4(MVD1)/UASp-shRNA(white)*. **(B)** SkpA is required for recruiting the atypical C(2)M cohesin complex along chromosome arms. Localization of the C(2)M subunit of the meiotic cohesin in control and *SkpA* RNAi germaria. GFP and C(3)G were immunostained in ovaries expressing GFP-C(2)M. C(2)M was diffused in *SkpA* RNAi meiotic cells. Scale bars = 2 μm. SkpA RNAi, *nos-GAL4(MVD1) UASp-GFP-c(2)M c(2)M⁺/UASp-shRNA(SkpA) c(2)M⁺*; control RNAi, *nos-GAL4(MVD1) UASp-GFP-c(2)M c(2)M⁺/UASp-shRNA(white) c(2)M⁺*. **(C)** The karyosome morphology in oocytes expressing shRNA against SCF subunits. Control RNAi, *nos-GAL4(MVD1)/UASp-shRNA(white)*. Cul1 RNAi; *nos-GAL4(MVD1)/UASp-shRNA(Cul1)*. Error bars represent SEM from triplicated experiments representing 47–66 oocytes. **(D)** The karyosome morphology in oocytes expressing shRNA against F-box proteins. Control RNAi, *nos-GAL4(MVD1)/UASp-shRNA(white)*; Fbxo42 RNAi, *nos-GAL4(MVD1)/UASp-shRNA(Fbxo42)*; Slmb RNAi, *nos-GAL4(MVD1)/UASp-shRNA(Slmb)*. Error bars represent SEM from triplicated experiments representing 21–40 oocytes. **(E)** The karyosome morphology in oocytes overexpressing GFP-Wrd. Wild-type, *nos-GAL4(MVD1) Wrd⁺/Wdr⁺*; GFP-Wrd, *nos-GAL4(MVD1) Wrd⁺/UASp-GFP-Wrd Wrd⁺*. Error bars represent SEM from triplicated experiments representing 29–36 oocytes. *, $P < 0.05$; **, $P < 0.01$; ***, $P < 0.001$.

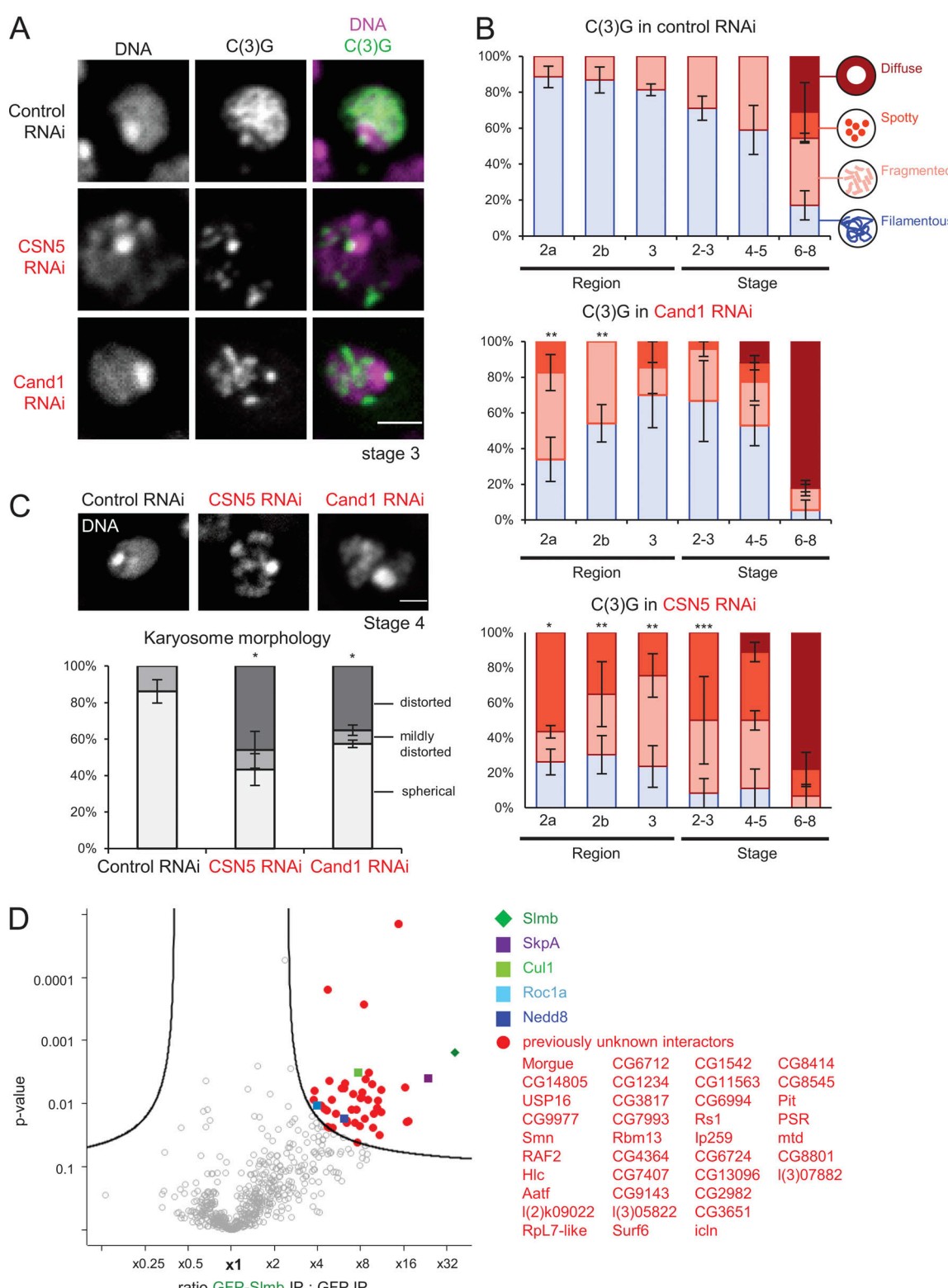

Figure S3. **Importance of the neddylation cycle and Slmb/βTrcp-interacting proteins. (A)** Localization of the synaptonemal complex component C(3)G in control, *CSN5* and *Cand1* RNAi. Scale bar = 2 µm. **(B)** C(3)G localization patterns in control, *Cand1*, and *CSN5* RNAi throughout meiotic progression. Error bars represent SEM from triplicated experiments representing 11–31 germaria/oocytes for each region/stage. **(C)** The karyosome morphology in oocytes expressing shRNA against *CSN5* and *Cand1*. Control RNAi, *nos-GAL4(MVD1)/UASp-shRNA(white)*; CSN5 RNAi, *nos-GAL4(MVD1)/UASp-shRNA(CNS5)*; Cand1 RNAi, *nos-GAL4(MVD1)/UASp-shRNA(Cand1)*. Error bars represent SEM from triplicated experiments representing 29–42 oocytes. Scale bar = 2 µm. **(D)** A volcano plot of proteins immunoprecipitated with GFP-Slmb/βTrcp or GFP from ovaries. For each protein, the mean ratio of signal intensities found in immunoprecipitates of GFP-Slmb/βTrcp and GFP alone was plotted against the statistical significance (the P value given by *t* test) from biological duplicates. *, P < 0.05; **, P < 0.01; ***, P < 0.001.

**Table S1 is provided online as a separate Excel file and lists phenotypes of lines expressing shRNA or a GFP-tagged protein.**

