## [Peer Review File · The Journal of Cell Biology]

SCF-Fbxo42 promotes synaptonemal complex assembly by downregulating PP2A-B56

Pedro Barbosa, Liudmila Zhaunova, Simona Debilio, Verdiana Steccanella, Van Kelly, Tony Ly, and Hiroyuki Ohkura

Corresponding Author(s): Hiroyuki Ohkura, University of Edinburgh

Review Timeline:

Submission Date:	2020-09-24
Editorial Decision:	2020-10-21
Revision Received:	2020-11-20
Editorial Decision:	2020-11-24
Revision Received:	2020-11-26

Monitoring Editor: Arshad Desai

Scientific Editor: Melina Casadio

Transaction Report:

DOI: <https://doi.org/10.1083/jcb.202009167>

October 21, 2020

Re: JCB manuscript #202009167

Prof. Hiroyuki Ohkura
University of Edinburgh
Wellcome Centre for Cell Biology
Michael Swann Building
Max Born Crescent
Edinburgh EH9 3BF
United Kingdom

Dear Prof. Ohkura,

Thank you for submitting your manuscript entitled "SCF-Fbxo42 promotes synaptonemal complex assembly by downregulating PP2A-B56". The manuscript was assessed by expert reviewers, whose comments are appended to this letter. We invite you to submit a revision if you can address the reviewers' key concerns, as outlined here.

You will see that the reviewers found the report of a role for the SCF complex in karyosome formation and SC formation/maintenance in the *Drosophila* ovary of interest. They felt that the data is of high quality and requested clarifications and edits/discussion that we feel will improve the manuscript. We recommend that you address the reviewers' remarks with appropriate text changes. We would be happy to further discuss the reviews if you anticipate any issues addressing them or have any questions.

GENERAL GUIDELINES:

Text limits: Character count for a Report is < 20,000, not including spaces. Count includes title page, abstract, introduction, results, discussion, acknowledgments, and figure legends. Count does not include materials and methods, references, tables, or supplemental legends.

Figures: Reports may have up to 5 main and up to 3 supplemental figures. Up to 10 supplemental videos or flash animations are allowed. A summary of all supplemental material should appear at the end of the Materials and methods section. To avoid delays in production, figures must be prepared according to the policies outlined in our Instructions to Authors, under Data Presentation, <https://jcb.rupress.org/site/misc/ifora.xhtml>. All figures in accepted manuscripts will be screened prior to publication.

As you may know, the typical timeframe for revisions is three to four months. However, we at JCB realize that the implementation of social distancing and shelter in place measures that limit spread of COVID-19 also pose challenges to scientific researchers. Lab closures especially are preventing scientists from conducting experiments to further their research. Therefore, JCB has waived the revision time limit. We recommend that you reach out to the editors once your lab has reopened to decide on an appropriate time frame for resubmission. Please note that papers are generally considered through only one revision cycle, so any revised manuscript will likely be either accepted or rejected.

Thank you for this interesting contribution to Journal of Cell Biology. You can contact us at the journal office with any questions, cellbio@rockefeller.edu or call (212) 327-8588.

Sincerely,

Arshad Desai, PhD
Editor, Journal of Cell Biology

Melina Casadio, PhD
Senior Scientific Editor, Journal of Cell Biology

Reviewer #1 (Comments to the Authors (Required)):

In this manuscript "SCF-Fbxo42 promotes synaptonemal complex assembly by downregulating PP2A-B56", Barbosa et al. demonstrate that SCF-Slmb (beta-TrCP) and SCF-Fbxo42 are required for the assembly and maintenance of the synaptonemal complex (SC) in the fly. Through an RNAi screen targeting ubiquitin-associated enzymes, the authors discovered that SCF components, SkpA and Cul1, are required for SC assembly. The authors further identified Slmb and Fbxo42 as two F-Box proteins that constitute the SCF required for SC regulation. They focused on Fbxo42 and identified PP2A subunits as Fbxo42-interacting proteins. It was further shown that the level of Wrd, a regulatory subunit of PP2A, is increased in the absence of Fbxo42 and that overexpression of PP2A indeed leads to SC defects, consistent with the idea that PP2A is an important substrate of SCF-Fbxo42, which has to be degraded to support proper SC assembly.

This is a straightforward paper with interesting novel discoveries showing the requirement of SCF in SC assembly in the fly. Due to the modular nature of the SCF complex and its diverse substrates and functions, it has been difficult to study how SCF controls specific cellular processes. Using biochemical purifications, the authors have identified not only two F-box proteins, but also a likely target of one of the F-box proteins. The data presented here support that Slmb and Fbxo42 are required for SC assembly and that regulating PP2A levels is essential for proper SC formation. Overall, this is an exciting discovery. However, it would have been great if the targets of SCF-Slmb have been explored further. The nature of SC defects in the SCF RNAi-treated animals and in animals over-expressing PP2A also need further characterization.

Here are specific points:

- 1) In Summary, it is noted that "Silva et al have found...". However, the first author's last name is Barbosa. Please resolve this discrepancy.
- 2) The first paragraph of the Results on page 5 is somewhat redundant with the Introduction.
- 3) The results in Figures S1A and S1B need to be fully explained in the main text. In particular, it has to be noted that gamma-H2Av was used as a marker for meiotic DSB formation (Figure S1A) and that Mnk is the homolog of Chk2 in *Drosophila* and has previously been implicated in the meiotic checkpoint (Abdu et al., 2002 *Curr Biol*).
- 4) While the evidence is clear in that SCF is required for the loading of C(G)3, it is not shown whether this is due to the failure in axis assembly and/or homolog pairing. Have the authors examined the meiotic cohesins or other SC components in the absence of SCF components? In Figure S2, it appears that C(2)M is found in both cytoplasm and nucleoplasm in SkpA RNAi-treated animals. Is it possible that SCF plays a role in the nuclear import of C(2)M?
- 5) When SC morphology is quantified (Figures 1D, 2E, 3B, and 4C), please consider showing representative immunofluorescence images of diffuse, spotty, fragmented, and filamentous C(3)G along with the cartoon diagrams to fully demonstrate the distinct RNAi phenotypes.
- 6) All graphs quantifying the SC morphology (Figures 1D, 2E, 3B, and 4C) include stages 4-5 and 6-8, which are not depicted in the diagram of fly ovaries shown in Figure 1E. I think showing the late prophase events in the cartoon (e.g. karyosome formation and SC disassembly) will greatly benefit the readers who may not be familiar with the meiotic progression in fly ovaries.
- 7) Can the authors comment on potential differences on SC morphology between SkpA and Cul1 RNAi-treated samples? It appears that Cul1 RNAi-treated animals were able to achieve close-to-normal C(3)G loading in the 2b region (Figures 1D and 2E), whereas SkpA RNAi-treated animals fail to do so.
- 8) On page 7, lines 169, please add commas when listing Cdc25A, Emi1, and Plk4.
- 9) Both 14-3-3 and PP2A were identified as binding partners of Fbxo42, and it was mentioned in the text that fly lines expressing GFP-tagged 14-3-3 have been generated (page 9, line 205). What were the results? Have the authors also tested the over-expression of 14-3-3 isoforms? Even if these did not result in SC defects, please describe these results.
- 10) I find that the Discussion of this paper is rather weak. The authors stated that the discovery demonstrating the involvement of SCF in SC regulation is novel and significant. While I generally agree with this claim, Jeremy Wang's group has recently shown that the mammalian SCF is important for SC maintenance (Guan et al., 2020), and it has to be incorporated into the Discussion.

There are many areas that can be discussed further in the Discussion. The SCF complex generally recognizes a phospho-degron to bind targets for polyubiquitination. What's known about the phosphorylation or ubiquitination of PP2A-B56 in fly or in other species? What is known about the regulation of PP2A-B56 activity? Even though the evidence is still sparse, it has now become clear that SC assembly/disassembly is regulated by meiotic kinases and phosphatases in diverse

eukaryotes. However, to my knowledge, it is the first time to see the involvement of PP2A-B56 in SC assembly.

Moreover, it would be nice to see some insights into how SCF-Slmb might regulate SC assembly. Finding Slmb as an F-box for SCF in SC regulation is an important part of this work. Therefore, it is disappointing to see that it is not discussed at all and is missing even in the model (Figure 4E). Have the authors tested the level of Wrd in Slmb RNAi-treated animals? Have the authors attempted to purify Slmb-interacting proteins?

11) For clarification, in the model shown in Figure 4E, the phosphorylation mark is on the chromosome axis and not on the central region. What do we know about phosphorylation of the SC in the fly?

Reviewer #2 (Comments to the Authors (Required)):

Barbosa et al provide evidence that the ubiquitin ligase complex SCF is important for karyosome formation and for the assembly and maintenance of the SC in the Drosophila ovary. The authors found that the SCF complex works through the F-box proteins Slmb and FBx042 proteins. The SCF complex and FBx042 downregulates the phosphatase subunit PP2A-B56 demonstrating that phosphatases play roles in the early Drosophila ovary.

The discovery of a ubiquitin ligase complex and the downstream PP2A phosphatase regulating karyosome formation and SC assembly in Drosophila ovary is interesting since the roles post-translational modification play in early meiotic events is still unclear. To put these roles in better perspective the paper needs some revisions and additional experiments to properly understand the true nature of the defects in these mutants.

The biggest issue is the way the paper is written. Readers may well come away with the impression that the SCF complexes is directly regulating SC assembly and disassembly. However, the strong karyosome defects shown in SkpA RNAi oocytes strongly suggest that the primary defect may well lie in chromatin organization/ karyosome structure. Unfortunately, karyosome defects are never mentioned beyond the initial characterization of SkpA. SC components would not be expected to be loaded or maintained on abnormal chromatin/ karyosome structure. While the model does show the phosphorylation on the DNA, this is not clearly explained in text or the figure legend. This indirect method of SC regulation needs more speculation. The karyosome defects that are evident in most of the images for all the RNAi lines tested are not mentioned except for skpA RNAi and this important phenotype is not included in the abstract.

Karyosome defect data needs to be provided for all the mutants where C(3)G localization is described. This will show if there is indeed a correlation between the level of karyosome defect and defects in the SC (as suggested by the images shown).

On a minor note the figure legends need more information on the transgenic strains used. For Figure S2 the authors give the impression they are staining with a C(2)M antibody. The methods reveal that a GFP antibody was used to detect an C(2)M-GFP overexpression construct. It is also unclear if the C(2)M-GFP is the only source of C(2)M or is untagged C(2)M is also present? The presence of untagged C(2)M would explain why it appears C(2)M is not loading but there are still tracts of C(3)G. In c(2)M mutants C(3)G fails to form any tracts and in ord mutants C(2)M is lost from the chromosomes arms with timing similar to when C(3)G is lost. It is hard to interpret why there would still be tracts of C(3)G if C(2)M is completely lost unless only a subset of C(2)M is being monitored. Clearer genotypes should be provided in all figure legends when a transgene has been

used.

Some minor notes:

The methods cite Zhaunova et al 2016 but it is not in the references.

In Figure S1 how are the meiotic cells identified? No mention is made of a SC marker or Orb.

In Figure S1 why do you think the RNAi resistant strain did not fully rescue? Inclusion of the full genotype might help explain this partial result.

Reviewer #3 (Comments to the Authors (Required)):

During meiosis, the pairing and alignment of homologous chromosomes is stabilized by the synaptonemal complex (SC), a higher order chromosome structure consisting of the juxtaposed protein axes of each homolog plus SC central region components. The SC has long been recognized as an important, conserved structure in meiosis, one that regulates many aspects of proper homologous recombination. However, many questions remain outstanding about how the SC is formed and then disassembled. This interesting paper uncovers a role for the ubiquitin ligase SCF in this process in *Drosophila melanogaster*, and further identifies specific Fbox versions of SCF important for this and identifies one important target, the protein phosphatase PP2A-B56. These are intriguing findings that will open up new lines of inquiry. The data are of high quality and the paper is clearly written and well reasoned.

Major point:

The abnormal karyosome phenotype is treated somewhat inconsistently and comes across as a bit of a distraction because it is introduced at the beginning of the results and then abandoned. It would be helpful to provide some discussion of the potential relationship of the karyosome defects to the earlier SC defects, and to provide explicit mention of karyosome phenotypes throughout the paper, not just at the beginning for the SkpA RNAi.

Minor points:

Fig 1B is not cited in the text; presumably this could go on line 106

Fig. S1B: mnk heterozygosity appears to make the effects of the SkpA RNAi worse. Was that reproducible? What are the implications of this?

Fig S2B: C(2)M levels look greatly elevated in addition to being diffuse. Is this correct, or are these not matched exposures between the RNAi and control images? Was that reproducible? This could be commented on in the manuscript.

Line 134: Fig 2C is cited out of order.

Lines 142-145: Did Cul1 RNAi cause similar karyosome defects as SkpA RNAi did?

Line 169: missing comma after Cdc25A

I would have found it useful if there were a discussion of the implications of there being two Fbox proteins that are non-redundantly required for SC formation.

I would have also found it useful if there were explicit statement about what can and cannot be

concluded from negative results of the overexpression experiments.

JCB manuscript #202009167

Dear Arshad and Melina

We are pleased to have received such positive and constructive comments from all the reviewers. Following your advice, we have incorporated all comments from the reviewers mainly by changing the text within the tight word limit, but also added some experimental data where appropriate, as detailed below. This revision has greatly improved the manuscript, which we believe is now suitable for publication in JCB.

With kind regards

Hiro

Hiro Ohkura

Professor of Cell Biology and Wellcome Investigator in Science
Wellcome Trust Centre for Cell Biology
The University of Edinburgh
Edinburgh EH9 3BF, UK
+44-131-650-7094
h.ohkura@ed.ac.uk

[Summary of revision]

New experimental data added

- Karyosome phenotypes in RNAi of Cul1, Slmb and Fbox42, and overexpression of GFP-Wrd have been included (both representative images and graphs) (Fig S2).

Minor additions/changes of the text

- Summary: Corrected from Silva to Barbosa.
- The first paragraph of Results & Discussion has been removed to avoid an overlap with Introduction.
- The last section of Results and Discussion has a fuller discussion including: underlying chromosome defects (karyosome, meiotic cohesin C(2)M). conservation in mammals by citing Guan et al 2000.
- Implication of non-redundancy between the two F-box proteins is discussed.
- All results of overexpression experiments are included with a comment on the limitation of interpretation from negative results.
- A comma has been added in Line 169.
- Fig 1B (now 1C) is cited in the text.
- Representative images of synaptonemal complex morphologies are included (Fig 1E).
- Later stages have been added in diagrams of meiotic progression (Fig 1A).
- Fuller descriptions of strain genotypes are provided in figure legends and Methods & Materials.
- All actual p-values are included in Methods & Materials.
- The marker used to identify meiotic cells has been specified in the legend of Fig S1.
- A general nature of the *mnk* mutation enhancing karyosome defects is mentioned in the legend of FigS1.
- Table S1 includes basic characterisation of all RNAi or overexpression phenotypes and is referred where appropriate.
- Zhaunova et al is included in the reference list.

Reviewer #1:

Overall, this is an exciting discovery.

We are very pleased to hear very positive comments and constructive suggestions which we have incorporated to the revised manuscript as follows.

it would have been great if the targets of SCF-Slmb have been explored further.

To identify the target of SCF-Slmb, we have done co-immunoprecipitation of Slmb followed by mass-spec, but did not find good candidates we wished to follow up. The data is attached below for inspection by the reviewers.

The nature of SC defects in the SCF RNAi-treated animals and in animals over-expressing PP2A also need further characterization.

Defects in the karyosome and recruitment of the C(2)M meiotic cohesin complex suggest defects in underlying chromosome organisation. This is briefly discussed in the last section of Results & Discussion. Additional data on the karyosome defects have been added in Fig. S2,S3.

1) In Summary, it is noted that "Silva et al have found...". However, the first author's last name is Barbosa. Please resolve this discrepancy.

Thank you for pointing this out. We have corrected it.

2) The first paragraph of the Results on page 5 is somewhat redundant with the Introduction.

It has been removed.

3) The results in Figures S1A and S1B need to be fully explained in the main text. In particular, it has to be noted that gamma-H2Av was used as a marker for meiotic DSB formation (Figure S1A) and that Mnk is the homolog of Chk2 in Drosophila and has previously been implicated in the meiotic checkpoint (Abdu et al., 2002 Curr Biol).

Additional information has been included in the main text.

*4) While the evidence is clear in that SCF is required for the loading of C(G)3, it is not shown whether this is due to the failure in axis assembly and/or homolog pairing. Have the authors examined the meiotic cohesins or other SC components in the absence of SCF components? *

We have looked at a subunit of the meiotic cohesin, C(2)M. It showed reduced signals on meiotic chromosomes in SkpA RNAi, suggesting defects in axis assembly. This is briefly discussed in the last section of Results & Discussion.

In Figure S2, it appears that C(2)M is found in both cytoplasm and nucleoplasm in SkpA RNAi-treated animals. Is it possible that SCF plays a role in the nuclear import of C(2)M?

It is a possibility, but it is also possible that chromosome association of C(2)M may be required to retain it in the nucleus.

5) When SC morphology is quantified (Figures 1D, 2E, 3B, and 4C), please consider showing representative immunofluorescence images of diffuse, spotty, fragmented, and filamentous C(3)G along with the cartoon diagrams to fully demonstrate the distinct RNAi phenotypes.

A representative image of each category of SC morphology has been included in Fig 1E.

6) All graphs quantifying the SC morphology (Figures 1D, 2E, 3B, and 4C) include stages 4-5 and 6-8, which are not depicted in the diagram of fly ovaries shown in Figure 1E. I think showing the late prophase events in the cartoon (e.g. karyosome formation and SC disassembly) will greatly benefit the readers who may not be familiar with the meiotic progression in fly ovaries.

We have included later stages of oogenesis in the diagram (Fig 1A).

7) Can the authors comment on potential differences on SC morphology between SkpA and Cul1 RNAi-treated samples? It appears that Cul1 RNAi-treated animals were able to achieve close-to-normal C(3)G loading in the 2b region (Figures 1D and 2E), whereas SkpA RNAi-treated animals fail to do so.

We have to use a weaker driver for Cul1 RNAi than the driver used for SkpA RNAi, as Cul1 RNAi using this stronger driver severely impairs ovary development. A weaker phenotype of Cul1 RNAi is likely due to weak knockdown caused by the use of a weaker driver of shRNA. The use of the weaker driver is stated in the figure legend (Fig 2).

8) On page 7, lines 169, please add commas when listing Cdc25A, Emi1, and Plk4.
Thank you for pointing this out. Corrected.

9) Both 14-3-3 and PP2A were identified as binding partners of Fbxo42, and it was mentioned in the text that fly lines expressing GFP-tagged 14-3-3 have been generated (page 9, line 205). What were the results? Have the authors also tested the over-expression of 14-3-3 isoforms? Even if these did not result in SC defects, please describe these results.

No SC defects were observed by expression of GFP-14-3-3 ϵ or ζ . These results were included in the original Supplemental Table, but a reference to the Table was mistakenly omitted from the main text. We have corrected this error.

10) I find that the Discussion of this paper is rather weak.

We could not include a full discussion as the Report format has a tight word limit and the Discussion needs to be combined with Results. Within these constraints, we have added more discussion in the last section of Results & Discussion.

The authors stated that the discovery demonstrating the involvement of SCF in SC regulation is novel and significant. While I generally agree with this claim, Jeremy Wang's group has recently shown that the mammalian SCF is important for SC maintenance (Guan et al., 2020), and it has to be incorporated into the Discussion.

Thank you very much for pointing this out. Embarrassingly, we were not aware of this paper. Although it is in spermatocytes, this suggests that the role of SCF may be conserved also in mammals. We have included this information and cited the paper in the last section of Results & Discussion.

There are many areas that can be discussed further in the Discussion. The SCF complex generally recognizes a phospho-degron to bind targets for polyubiquitination. What's known about the phosphorylation or ubiquitination of PP2A-B56 in fly or in other species? What is known about the regulation of PP2A-B56 activity?

Recognition of the phospho-degron depends on specific F-box proteins (Slmb, Cdc4), rather than a general nature of SCF. There is no information about FBxo42 yet with regards to this. We are not aware of any report showing that PP2A-B56 is regulated by phosphorylation or ubiquitination.

Moreover, it would be nice to see some insights into how SCF-Slmb might regulate SC assembly. Finding Slmb as an F-box for SCF in SC regulation is an important part of this work. Therefore, it is disappointing to see that it is not discussed at all and is missing even in the model (Figure 4E). Have the authors tested the level of Wrd in Slmb RNAi-treated animals? Have the authors attempted to purify Slmb-interacting proteins?
We have observed the GFP-Wrd level in Slmb RNAi, and saw some decreases. However, it is difficult to draw conclusions, as Slmb RNAi affects the morphology/organisation of the germarium. To identify targets of SCF-Slmb, we have done co-immunoprecipitation of Slmb followed by mass-spec, but did not find good candidates we wished to follow up. A difficulty in studying Slmb is that it is involved in other processes during oogenesis, in contrast to Fbxo42 which seems to have a more specific role. The IP/MS data is attached below for inspection by the reviewers.

11) For clarification, in the model shown in Figure 4E, the phosphorylation mark is on the chromosome axis and not on the central region. What do we know about phosphorylation of the SC in the fly?

Although evidence suggest roles of phosphorylation on SC regulation, specific phosphorylations for this have not been identified. As SkpA RNAi affects chromosome localisation of the meiotic cohesin subunit C(2)M, we suspect that chromosome axis assembly is defective. Discussion of this has been included in the last section in Results & Discussion.

Reviewer #2:

The discovery ... is interesting since the roles post-translational modification play in early meiotic events is still unclear... To put these roles in better perspective the paper needs some revisions and additional experiments to properly understand the true nature of the defects in these mutants.

We are very happy to hear favourable comments and constructive suggestions. We have revised the manuscript accordingly as follows.

The biggest issue is the way the paper is written. the primary defect I may well lie in chromatin organization/ karyosome structure.

We have added brief discussion in the last section of Results and Discussion.

Karyosome defect data needs to be provided for all the mutants where C(3)G localization is described.

We have included data showing karyosome phenotypes (both representative images and graphs from triplicated experiments.) in Fig S2, S3.

On a minor note the figure legends need more information on the transgenic strains used. ... Clearer genotypes should be provided in all figure legends when a transgene has been used."

In the original manuscript, the figure legends were kept minimum to fit in tight word limits of the *Report* format, although Methods & Materials included full information. We have added detailed information on the strains to the legends of Fig 1, 4, S1, S2.

Some minor notes:

The methods cite Zhaunova et al 2016 but it is not in the references.

Thank you for pointing this out. We have added it to the reference list.

In Figure S1 how are the meiotic cells identified? No mention is made of a SC marker or Orb.

They are identified by an SC marker, C(3)G. This information has been included in the legends.

In Figure S1 why do you think the RNAi resistant strain did not fully rescue? Inclusion of the full genotype might help explain this partial result.

The RNAi-resistant transgene is under the control of the ubiquitin promoter, not the SkpA promoter. This may explain imperfect rescue of the phenotype. This information has been included in the figure legends.

Reviewer #3:

This interesting paper uncovers a role for the ubiquitin ligase SCF ... These are intriguing findings that will open up new lines of inquiry. The data are of high quality and the paper is clearly written and well reasoned.

We are delighted to hear positive comments. We have revised the manuscript to incorporate all suggestions.

Major point:

It would be helpful to provide some discussion of the potential relationship of the karyosome defects to the earlier SC defects, and to provide explicit mention of karyosome phenotypes throughout the paper, not just at the beginning for the SkpA RNAi.

We have added data showing karyosome phenotypes (both representative images and graphs from triplicated experiments.) in Fig S2, S3, and brief discussion has been added to the final section of Results and Discussion.

Minor points:

Fig 1B is not cited in the text; presumably this could go on line 106

Thank you. It is now cited as Fig 1C.

Fig. S1B: mnk heterozygosity appears to make the effects of the SkpA RNAi worse. Was that reproducible? What are the implications of this?

It is reproducible, but this enhancement is a fairly common phenomenon not specific to SkpA or SCF. mnk heterozygosity commonly enhances karyosome defects caused by RNAi of other unrelated genes. We have included this information in the figure legends.

Fig S2B: C(2)M levels look greatly elevated in addition to being diffuse. Is this correct, or are these not matched exposures between the RNAi and control images? Was that reproducible? This could be commented on in the manuscript.

These images were captured and processed under the identical conditions, but it is hard to draw conclusions from the immunostaining data, as proteins on chromosomes tend to be more difficult to stain, probably due to low antibody accessibility.

Line 134: Fig 2C is cited out of order.

This is not ideal, but helps to use the figure space efficiently.

Lines 142-145: Did Cul1 RNAi cause similar karyosome defects as SkpA RNAi did?

Yes, Cul1 RNAi causes similar karyosome defects. Data of the karyosome phenotype has been added in Fig S2, S3.

Line 169: missing comma after Cdc25A

Thank you. We have corrected.

I would have found it useful if there were a discussion of the implications of there being two Fbox proteins that are non-redundantly required for SC formation.

Non-redundancy and implication have now been discussed.

I would have also found it useful if there were explicit statement about what can and cannot be concluded from negative results of the overexpression experiments.

We added the statement as requested.

November 24, 2020

RE: JCB Manuscript #202009167R

Prof. Hiroyuki Ohkura
University of Edinburgh
Wellcome Centre for Cell Biology
Michael Swann Building
Max Born Crescent
Edinburgh EH9 3BF
United Kingdom

Dear Prof. Ohkura,

Thank you for submitting your revised manuscript entitled "SCF-Fbxo42 promotes synaptonemal complex assembly by downregulating PP2A-B56". Thank you for adding the karyosome phenotypic characterizations requested by the reviewers and for your responses to their comments. We feel that the revision is now stronger and appreciate your efforts to bolster the analyses and conclusions. We also feel that the Slmb MS data are valuable and likely to be informative for experts studying Slmb, even if the list did not yield strong candidates for your particular purpose here. We therefore recommend that you move the full MS dataset to the supplement and mention it in the text of the paper. We are happy to accommodate a reasonable extension of the character count to ensure you have space to discuss the results as needed. We would be happy to publish your paper in JCB pending the addition of these data and final revisions necessary to meet our formatting guidelines (see details below).

- 1) JCB Reports may have up to 5 main and 3 supplementary figures. There is space in the main figures to add data as needed and still meet this limit. Figures can span a full page as long as all panels fit on the page.
- 2) Statistical analysis: Error bars on graphic representations of numerical data must be clearly described in the figure legend. The number of independent data points (n) represented in a graph must be indicated in the legend. Statistical methods should be explained in full in the materials and methods. For figures presenting pooled data the statistical measure should be defined in the figure legends.
- 3) Materials and methods: Should be comprehensive and not simply reference a previous publication for details on how an experiment was performed. Please provide full descriptions in the text for readers who may not have access to referenced manuscripts.
 - For all cell lines, vectors, constructs/cDNAs, Drosophila lines, etc. - all genetic material: please include database / vendor ID (e.g., Addgene, ATCC, FlyBase, BDSC, etc.) or if unavailable, please briefly describe their basic genetic features *even if described in other published work or gifted to you by other investigators*
 - Please include species and source for all antibodies, including secondary, as well as catalog

numbers/vendor identifiers if available.

- Sequences should be provided for all oligos: primers, si/shRNA, gRNAs, etc.
- Microscope image acquisition: The following information must be provided about the acquisition and processing of images:
 - a. Make and model of microscope
 - b. Type, magnification, and numerical aperture of the objective lenses
 - c. Temperature
 - d. imaging medium
 - e. Fluorochromes
 - f. Camera make and model
 - g. Acquisition software
 - h. Any software used for image processing subsequent to data acquisition. Please include details and types of operations involved (e.g., type of deconvolution, 3D reconstitutions, surface or volume rendering, gamma adjustments, etc.).

4) A summary paragraph of all supplemental material should appear at the end of the Materials and methods section.

- Please include one brief descriptive sentence per item.

A. MANUSCRIPT ORGANIZATION AND FORMATTING:

Full guidelines are available on our Instructions for Authors page, <https://jcb.rupress.org/submission-guidelines#revised>. **Submission of a paper that does not conform to JCB guidelines will delay the acceptance of your manuscript.**

B. FINAL FILES:

-- High-resolution figure and video files: See our detailed guidelines for preparing your production-ready images, <https://jcb.rupress.org/fig-vid-guidelines>.

Thank you for this interesting contribution, we look forward to publishing your paper in the Journal of Cell Biology.

Sincerely,

Arshad Desai, PhD
Editor, Journal of Cell Biology

Melina Casadio, PhD
Senior Scientific Editor, Journal of Cell Biology